# Boosting Vision-and-Language Navigation in Urban Environments with a Hierarchical Spatial-Cognition Memory System

## Abstract

Vision-and-Language Navigation (VLN) in large-scale urban environments requires embodied agents to ground linguistic instructions in complex scenes and recall relevant experiences over extended time horizons. Prior modular pipelines offer interpretability but lack unified memory, while end-to-end (M)LLM agents excel at fusing vision and language yet remain constrained by fixed context windows and implicit spatial reasoning. We introduce **Mem4Nav**, a hierarchical spatial–cognition memory system that can augment most of the VLN backbones. Mem4Nav fuses a sparse octree for fine-grained voxel indexing with a semantic topology graph for high-level landmark connectivity, storing environmental context in trainable memory tokens embedded via a reversible Transformer. Long-term memory (LTM) losslessly compresses historical observations, while short-term memory (STM) caches recent entries for real-time local planning. At each step, the agent dynamically retrieves from STM for immediate context or queries LTM to reconstruct deep history as needed. When evaluated on the Touchdown and Map2Seq benchmarks, Mem4Nav demonstrates substantial performance gains across three distinct backbones (modular, LLM-based, and MLLM-based). Our method improves Task Completion by up to 13.3 percentage points and enhances path fidelity (nDTW) by more than 12 percentage points, while also reducing the final goal distance. Extensive ablation studies confirm the indispensability of both the hierarchical map and the dual memory modules. Our code is open-sourced via `https://anonymous.4open.science/r/anonymous_Mem4Nav-62B0/`.

## 1 Introduction

Vision-and-Language Navigation (VLN) requires an agent to follow free-form natural language instructions and navigate through complex visual environments to reach a specified target (Anderson et al., 2018; Gu et al., 2022). Most existing methods primarily address indoor VLN. One class of methods (Anderson et al., 2018; Chen et al., 2022; Kurita & Cho, 2020; Gao et al., 2023; Chen et al., 2024; Huo et al., 2023) frames the task as traversal on a discrete topological graph, allowing agents to teleport between fixed nodes without modeling motion uncertainty, which limits their applicability in real-world continuous spaces. Other techniques remove the reliance on such graphs by learning end-to-end action policies (Krantz et al., 2020; Chen et al., 2021; Raychaudhuri et al., 2021) or by predicting intermediate waypoints (Hong et al., 2022; An et al., 2024; Wang et al., 2024b). Action-based methods struggle with diverse semantic variations in scenes, while waypoint-based approaches do not generalize well to expansive outdoor settings. Recent work has attempted to extend VLN from indoor settings to outdoor urban environments(Schumann et al., 2024; Liu et al., 2024; Xu et al., 2025; Feng et al., 2024), yet it still lacks the ability to sustain long-term perception, memory, and autonomous decision-making over complex 3D scenes at city scale.

Recent VLN approaches fall into two camps. On one hand, **Hierarchical Modular Pipelines** decouple perception, mapping, planning and control, offering interpretability but relying on hand-crafted interfaces and lacking unified memory (Raychaudhuri et al., 2021; Hong et al., 2022; Du et al., 2025). On the other hand, **(M)LLM-Based Agents** leverage large (multimodal) language models to fuse vision and language, achieving near end-to-end performance but still bounded by fixed context windows and implicit spatial memory (Shah et al., 2023; Schumann et al., 2024; Liu et al., 2024; Xu et al., 2025). Neither paradigm natively supports efficient, lossless storage and retrieval of large-scale

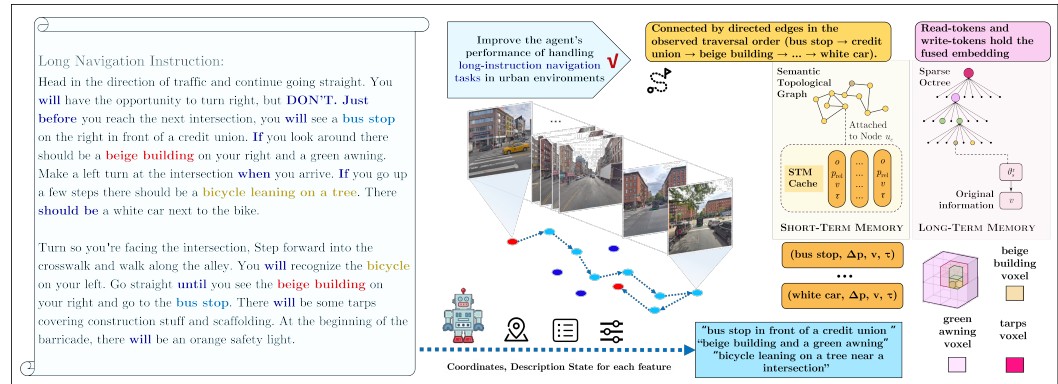

Figure 1: Long-instruction navigation in urban environments demands that agents retain both fine-grained spatial detail and high-level landmark semantics over many steps—a core challenge that leads to information loss or retrieval overload. Mem4Nav meets this by building a hierarchical spatial-cognition long-short memory system.

3D structure nor fast adaptation to dynamic, local changes. The primary bottleneck in urban VLN may be the agent's inability to model its current 3D spatial information, store it in memory in a structured form, and retrieve it quickly and efficiently when required. Therefore, based on existing research, we propose the following hypothesis: **the key to** endowing an embodied agent with complex autonomous decision-making capabilities in urban environments-and thus achieving more powerful Vision-and-Language Navigation is a **high-performance memory system** that is seamlessly integrated into the agent's other cognitive functions, such as perception and decision-making.

To bridge this gap, we propose **Mem4Nav**, a hierarchical 3D spatial–cognition long–short memory framework that augments any VLN backbone. After the visual encoder, we build a **sparse octree** for voxel-level indexing of observations, a **semantic topology graph** linking landmark nodes and intersections, a **long-term memory** reversible memory tokens and a compact **short-term memory cache** of recent entries in local coordinates for rapid adaptation. We evaluate Mem4Nav on two street-view VLN benchmarks, Touchdown (Chen et al., 2019) and Map2Seq (Schumann & Riezler, 2021), and use three backbones: a non-end-to-end modular pipeline, a prompt-based LLM navigation agent (Schumann et al., 2024), and a strided-attention MLLM navigation agent (Xu et al., 2025). Under the same training cost and hardware budget as the strongest baselines, Mem4Nav delivers absolute improvements of seven to thirteen percentage points in Task Completion, reduces the final stop distance by up to 1.6 m, and increases normalized DTW by more than ten percentage points. Ablation studies confirm that each component—the sparse octree, the semantic graph, the long-term memory tokens, and the short-term cache—is essential to these gains.

In summary, our contributions are:

- We introduce a dual-structured 3D map combining sparse octree indexing with a semantic topology graph, unifying fine-grained geometry and landmark connectivity.
- We design a reversible Transformer memory that losslessly compresses and retrieves spatially anchored observations at both octree leaves and graph nodes. We develop a short-term memory cache for high-frequency local lookups, and a unified retrieval mechanism that dynamically balances short- and long-term memories within the agent's attention.
- We demonstrate that Mem4Nav consistently enhances three distinct VLN backbones on Touchdown and Map2Seq, delivering substantial improvements in success rate, path fidelity, and distance metrics.

## 2 RELATED WORK

**Vision-and-Language Navigation (VLN).** VLN, introduced with the R2R benchmark (Anderson et al., 2018), requires agents to follow natural language instructions in visual environments. While numerous methods have addressed indoor navigation (Shridhar et al., 2020; Gao et al., 2023; Huo et al., 2023; Chen et al., 2023; Zheng et al., 2024a; Dai et al., 2023; Li et al., 2023; Chen et al., 2022; Guhur et al., 2021; Qi et al., 2021; Zhou et al., 2024b; Chen et al., 2024; Zhou et al., 2024a; An et al., 2024), outdoor urban settings present unique challenges due to their scale and complexity.

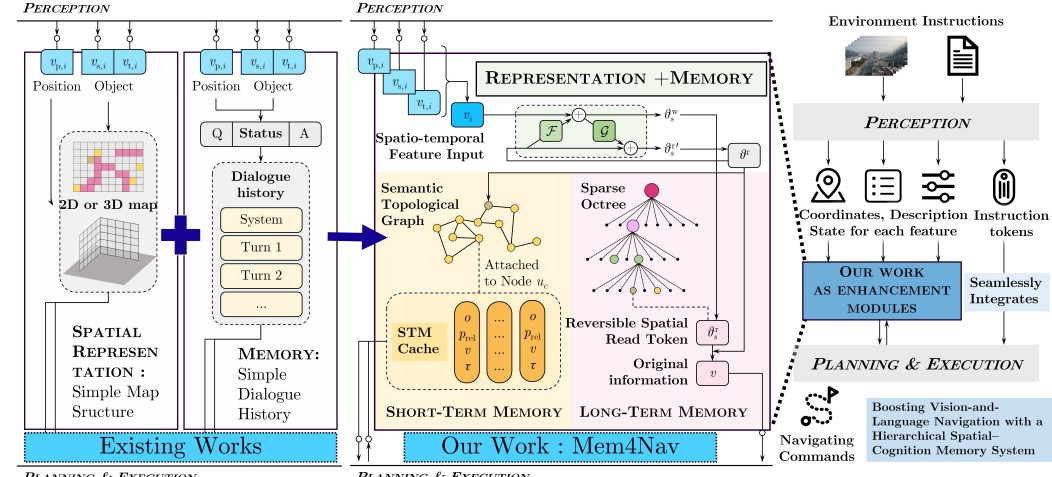

Figure 2: **Contributions of Mem4Nav**: Prior VLN systems treat spatial maps and memory as separate, using flat, monolithic maps that are either too detailed (noisy, slow to query) or too coarse (lossy), and simple text-based memory that merely appends raw history to instructions, leading to clutter and forgetting. Mem4Nav jointly implements a hierarchical spatial representation and a dual long–short memory mechanism, and can be seamlessly integrated into existing vision-and-language navigation pipelines to boost performance.

Datasets like Touchdown (Chen et al., 2019) and Map2Seq (Schumann & Riezler, 2021) were created to address this gap. Despite recent progress in outdoor VLN (Schumann et al., 2024; Liu et al., 2023; Xu et al., 2025; Gao et al., 2024; Wang et al., 2024a; Tian et al., 2024; Feng et al., 2024; 2025), efficiently handling long-horizon tasks remains a key challenge.

**Modular vs. End-to-End VLN Architectures.** VLN architectures typically fall into two paradigms. **Modular pipelines** decompose navigation into stages like perception and planning (Mirowski et al., 2018; Parvaneh et al., 2020), offering interpretability but often struggling with long-horizon consistency. In contrast, **end-to-end models** learn a direct mapping from multimodal inputs to actions. These range from early cross-modal Transformers (Schumann & Riezler, 2022) to modern agents based on Large Language Models (LLMs) (Qiao et al., 2023; Zhang et al., 2024a; Schumann et al., 2024; Zhou et al., 2024b) and Multimodal LLMs (MLLMs) (Xu et al., 2025). While simpler to train, they are constrained by fixed-size context windows and lack explicit spatial memory. Our work is motivated by the need for a structured, long-term memory system that complements both approaches.

**Spatial Representation Methods in VLN.** Effective VLN relies on robust spatial representations. Common approaches include point clouds, which offer geometric flexibility but can be slow to index (Wang et al., 2024b), and topological graphs, which abstract environments for efficient, high-level planning (Wang et al., 2025; Zemskova & Yudin, 2024). Mem4Nav integrates the strengths of both, using a sparse octree for fine-grained indexing and a landmark graph for macroscopic routing, creating a unified and hierarchical 3D representation.

**Memory Mechanisms in VLN.** Memory is crucial for grounding decisions in past experiences. While simple caches can aid in landmark re-recognition (Sun et al., 2025), recent methods for longer-horizon memory often inject historical text directly into a model's prompt. This strategy scales poorly and lacks structured spatial grounding in complex urban environments (Zeng et al., 2024). Mem4Nav addresses this limitation by embedding reversible memory tokens directly into its spatial representation, enabling efficient and structured recall over extended trajectories.

## 3 METHODOLOGY

Our proposed framework, **Mem4Nav**, enhances large-scale urban VLN by integrating a hierarchical spatial representation with a dual long-short memory system. This section details the core components of our architecture, focusing on its multi-level environmental mapping and memory management mechanisms.

## 3.1 HIERARCHICAL SPATIAL REPRESENTATION

To support both fine-grained geometric lookup and high-level route planning, Mem4Nav builds a dual-component spatial map. It uses a **sparse octree** for efficient, voxel-level indexing of the local 3D environment, complemented by a **semantic topological graph** that abstracts the world into key landmarks and their connections for macroscopic planning.

**Sparse Octree Indexing**

We discretize the continuous 3D space into a hierarchical octree of maximum depth $\Lambda$, where each level $\ell \in \{0, \ldots, \Lambda\}$ corresponds to axis-aligned cubes of side length $L/2^\ell$. Only those leaf cubes that the agent visits or that contain relevant observations are instantiated and stored in a hash map, ensuring both sparsity and $O(1)$ average lookup time. To recover 3D structure from RGB panoramas, we employ the universal monocular metric depth estimator UniDepth. (Piccinelli et al., 2024)

**Morton Code Addressing.** The agent's position $p_t = (x_t, y_t, z_t)$ is quantized to integer indices

$$\bar{p}_t = \left( \lfloor x_t \, 2^\Lambda/L \rfloor, \; \lfloor y_t \, 2^\Lambda/L \rfloor, \; \lfloor z_t \, 2^\Lambda/L \rfloor \right) \in \{0, \ldots, 2^\Lambda - 1\}^3,$$

which are interleaved to form a Morton code

$$\kappa(p_t) \; = \; \text{InterleaveBits}(\bar{p}_t) \; \in \; \{0, \ldots, 2^{3\Lambda} - 1\}.$$

This single integer uniquely identifies the visited leaf. On each visit, if $\kappa(p_t)$ is not already present, a new leaf entry is created; otherwise, the existing leaf's embedding is updated with the latest observation in constant time.

**Leaf Embedding Updates.** Each instantiated leaf maintains an aggregated embedding of the observations within its cube. Upon revisiting, the current feature vector $v_t$ is fused into this embedding via a reversible update operator, preserving both efficiency and information fidelity. More details are provided in appendix A.2.

### 3.1.1 SEMANTIC TOPOLOGICAL GRAPH

While the octree captures raw geometry, high-level navigation relies on semantic landmarks and decision points. We therefore maintain a dynamic directed graph $\mathcal{G} = (\mathcal{V}, \mathcal{E})$, where each node $u \in \mathcal{V}$ corresponds to a landmark or intersection and edges $(u_i, u_j) \in \mathcal{E}$ encode traversability and cost.

**Node Creation.** Given the current embedding $v_t$ and existing node descriptors $\{\phi(u)\}$, we create a new node whenever

$$\min_{u \in \mathcal{V}} \|v_t - \phi(u)\| > \delta,$$

assigning the new node the position $p_t$ and initializing its descriptor to $v_t$.

**Edge Weighting.** Whenever the agent moves from node $u_{t-1}$ to $u_t$, we add or update the directed edge $(u_{t-1}, u_t)$ with weight

$$w_{t-1,t} \; = \; \alpha \, \|p_{t-1} - p_t\|_2 \; + \; \beta \, c_{\text{instr}},$$

where $c_{\text{instr}}$ encodes instruction-based penalties (e.g. turns). If the edge already exists, its weight is averaged to smooth out noise.

**Query Modes.** At decision time, the agent may perform:

- **Voxel lookup**: compute $\kappa(p)$ and fetch the corresponding octree leaf embedding for precise local reasoning.
- **Graph lookup**: run a shortest-path algorithm on $\mathcal{G}$ to retrieve a sequence of landmark nodes for macro-scale routing.

**Combined Query Modes.** At query time, the agent can:

- **Voxel lookup**: given a precise coordinate, compute $\kappa$ and fetch $\theta_\kappa^r$.
- **Node lookup**: given a semantic goal node $u_g$, perform a shortest–path search (e.g. Dijkstra) on $\mathcal{G}$ to retrieve the sequence of graph tokens along the plan.

This dual representation ensures that Mem4Nav can rapidly retrieve the memory tokens most relevant to either microscopic obstacle avoidance or macroscopic route guidance, all within real–time constraints.

Figure 3: **The Mem4Nav Pipeline.** Perceived observations are encoded into a dual memory system. Long-Term Memory uses a reversible Transformer to losslessly store information in a hierarchical map composed of a sparse octree and a semantic graph. Short-Term Memory caches recent observations for fast, local lookups. During planning, the agent queries STM for immediate context, falling back to a search over LTM for deeper history. The retrieved memory vectors are then fused with current perception to guide the agent's actions.

## 3.2 LONG–TERM MEMORY WITH REVERSIBLE TOKENS

Long–Term Memory (LTM) provides high–capacity, lossless storage of spatially anchored observations via virtual memory tokens embedded in both octree leaves and semantic graph nodes. Each spatial element $s$ (leaf or node) maintains a read–token $\theta_s^r$ and write–token $\theta_s^w$, both in $\mathbb{R}^d$. New observations $v_t \in \mathbb{R}^d$ are absorbed into LTM by a bijective update, and past information can be exactly reconstructed when needed.

**Reversible Transformer Block.** We adopt a reversible architecture $\mathcal{R}$ composed of $L$ layers. At each layer $\ell$, inputs $(x_\ell^1, x_\ell^2)$ are transformed via two submodules $F_\ell$ and $G_\ell$:

$$y_\ell^1 = x_\ell^1 + F_\ell(x_\ell^2), \qquad y_\ell^2 = x_\ell^2 + G_\ell(y_\ell^1),$$
$$x_\ell^2 = y_\ell^2 - G_\ell(y_\ell^1), \qquad x_\ell^1 = y_\ell^1 - F_\ell(x_\ell^2).$$

Here each $F_\ell, G_\ell$ is a lightweight adapter atop a frozen Transformer layer. Collectively, $\mathcal{R}$ maps $(\theta_s^r, v_t) \mapsto \theta_s^w$ and supports exact inverse.

**Write Update.** When an observation $v_t$ falls into spatial element $s$:

$$\theta_s^w \leftarrow \mathcal{R}\big(\theta_s^r \,\|\, v_t\big), \qquad \theta_s^r \leftarrow \theta_s^w.$$

Concatenation $\|$ yields a $2d$-dimensional input. Because $\mathcal{R}$ is bijective, no information is lost: the original $(\theta_s^r, v_t)$ can be recovered by the inverse pass.

**Cycle–Consistency Training.** To enforce faithful reconstruction, we minimize a cycle consistency loss on synthetic trajectories:

$$\mathcal{L}_{\text{cycle}} = \mathbb{E}_v\Big[\big\|v - \widehat{v}\big\|_2^2\Big], \quad \widehat{v} = \pi_v\Big(\mathcal{R}^{-1}\big(\mathcal{R}(\theta^r; v)\big)\Big),$$

where $\pi_v$ is a small decoder projecting reversed hidden states back to the embedding space. Jointly with any downstream navigation loss, this trains the reversible block to faithfully encode and decode.

**Retrieval from LTM.** At decision time, if local cache misses, we compose a query $q_t = \text{Proj}([v_t; p_t])$ and perform an approximate nearest neighbor lookup over $\{\theta_s^r\}$ using HNSW (Hierarchical Navigable Small World, A.2.3) graphs. For each retrieved token $\theta_{s_i}^r$, we recover the original embedding via inverse transform:

$$\widehat{v}_{s_i} = \mathcal{R}^{-1}\big(\theta_{s_i}^r\big),$$

and then decode:

$$\widehat{p}_{s_i} = \pi_p(\widehat{v}_{s_i}), \quad \widehat{d}_{s_i} = \pi_d(\widehat{v}_{s_i}),$$

where $\pi_p$, $\pi_d$ are MLP decoders for position and descriptor. A small set of top-$m$ memories $\{(\widehat{p}_{s_i}, \widehat{d}_{s_i})\}$ is fed into the policy for global reasoning.

## 3.3 Short–Term Memory Cache

Short–Term Memory (STM) is a fixed–size, high–frequency buffer attached to the current semantic node $u_c$. It stores the most recent observations in relative coordinates for rapid local lookup and dynamic obstacle avoidance.

**Entry Structure.** Each STM entry $e = (o,\ p_{\text{rel}},\ v,\ \tau)$ comprises:

- $o$: object or event identifier (e.g. car, traffic_light),
- $p_{\text{rel}} = p_t - p_{u_c}$: coordinate relative to current node,
- $v \in \mathbb{R}^d$: multimodal embedding,
- $\tau$: timestamp or step index.

**Replacement Policy.** To maximize hit rate under capacity $K$, we combine frequency and recency:
$$\text{Score}(e_i) = \lambda \, \text{freq}(e_i) \ - \ (1 - \lambda)\left(t_{\text{now}} - \tau_i\right),$$
where $\text{freq}(e_i)$ is the access count. On cache full and new entry:
$$e_{\text{evict}} = \arg\min_i \text{Score}(e_i).$$

This Frequency–and–Least–Frequently Used policy preserves both frequently accessed and recently used items.

**STM Retrieval.** At time $t$, given current embedding $v_t$ and relative query $q_{\text{rel}}$:
$$\mathcal{C} = \{\, e_i : \|p_{\text{rel},i} - q_{\text{rel}}\| \le \epsilon \},$$
then compute cosine similarity
$$s_i = \frac{\langle v_t,\ v_i \rangle}{\|v_t\|\|v_i\|}, \quad i \in \mathcal{C},$$
and return top–$k$ entries $\{e_{i_1}, \ldots, e_{i_k}\}$. Both filtering and similarity ranking cost $O(K)$, with $K \le 128$ in practice.

By combining LTM for deep history and STM for immediate context, our Mem4Nav system achieves both large–scale recall and rapid local adaptation in real time.

## 3.4 Multi–Level Memory Retrieval and Decision Making

At each time step $t$, with current observation embedding $v_t$ and position $p_t$, Mem4Nav first attempts a short-term memory lookup by computing the relative query $q_{\text{rel}} = p_t - p_{u_c}$, filtering STM entries within radius $\epsilon$, and ranking them by cosine similarity. If the highest similarity exceeds threshold $\tau$, the agent aggregates the top-$k$ STM embeddings into $m_{\text{STM}}$; otherwise it falls back to long-term memory by projecting $q_t = \text{Proj}([v_t; p_t])$, performing an HNSW search over all read-tokens $\{\theta_s^r\}$ in the sparse octree and semantic graph, decoding the top-$m$ tokens via the reversible Transformer inverse into $\{\widehat{v}_{s_i}\}$, and aggregating them into $m_{\text{LTM}}$. The final memory vector is chosen as
$$m_t = \begin{cases} m_{\text{STM}}, & \max_i \langle v_t, v_i \rangle \ge \tau, \\ m_{\text{LTM}}, & \text{otherwise}, \end{cases}$$
which is concatenated to the baseline keys and values $\{K, V\}$ in the policy's cross-attention:
$$K' = [K; m_t], \quad V' = [V; m_t],$$
and combined via a learned gate $\alpha_t$:
$$\text{Out}_t = \alpha_t \, \text{Attn}(Q, K', V') + (1 - \alpha_t) \, \text{Attn}(Q, K, V).$$

The result then flows through the feed-forward and action-selection layers, allowing the agent to rely on fresh local context whenever possible and deeper historical cues when necessary.

## 4 Experiments

We evaluate Mem4Nav on two large-scale urban VLN benchmarks, Touchdown (Chen et al., 2019) and Map2Seq (Schumann & Riezler, 2021). To demonstrate its broad applicability, our system is tested by augmenting three distinct backbones: a modular pipeline, the LLM-based agent VELMA (Schumann et al., 2024), and the MLLM-based agent FLAME (Xu et al., 2025).

Table 1: Test-set performance on Touchdown and Map2Seq backbones, with ("+Mem4Nav") and without memory.

| Model | Touchdown Dev | | | Touchdown Test | | | Map2Seq Dev | | | Map2Seq Test | | |
|---|---|---|---|---|---|---|---|---|---|---|---|---|
| | TC↑ | SPD↓ | nDTW↑ | TC↑ | SPD↓ | nDTW↑ | TC↑ | SPD↓ | nDTW↑ | TC↑ | SPD↓ | nDTW↑ |
| VLN-Trans (Majumdar et al., 2021) | 15.00 | 20.30 | 27.00 | 16.20 | 20.80 | 27.80 | 18.60 | — | 31.10 | 17.00 | — | 29.50 |
| ARC+L2S (2020) (Cho et al., 2020) | 19.48 | 17.05 | — | 16.68 | 18.84 | — | — | — | — | — | — | — |
| ORAR (2022) (Liu et al., 2022) | 30.05 | 11.12 | 45.50 | 29.60 | 11.79 | 45.30 | 49.88 | 5.87 | 62.70 | 47.75 | 6.53 | 62.10 |
| VLN-Video (2024) (Zhang et al., 2024b) | 34.50 | 9.60 | — | 31.70 | 11.20 | — | — | — | — | — | — | — |
| Loc4Plan (2024) (Tian et al., 2024) | 34.50 | 10.50 | — | 32.90 | 11.50 | — | 48.00 | 7.00 | — | 45.30 | 7.20 | — |
| Hierarchical Modular Pipeline | 31.93 | 12.84 | 46.07 | 29.27 | 13.05 | 44.29 | 53.03 | 6.22 | 69.06 | 50.54 | 6.33 | 65.50 |
| + Mem4Nav (ours) | **45.18** | **11.21** | **59.03** | **42.21** | **11.95** | **56.36** | **58.19** | **5.49** | **74.74** | **57.64** | **5.54** | **73.57** |
| VELMA Baseline (Schumann et al., 2024) | 29.83 | 14.67 | 43.44 | 27.38 | 15.03 | 41.93 | 52.75 | 6.78 | 66.45 | 48.70 | 6.80 | 62.37 |
| + Mem4Nav (ours) | **35.29** | **12.16** | **55.35** | **34.04** | **12.90** | **48.82** | **58.33** | **6.01** | **75.06** | **56.84** | **6.10** | **72.71** |
| FLAME Baseline (Xu et al., 2025) | 41.28 | 9.14 | 55.96 | 40.20 | 9.53 | 54.56 | 56.95 | 5.95 | 71.36 | 52.44 | 5.91 | 67.72 |
| + Mem4Nav (ours) | **50.10** | **9.01** | **65.05** | **48.48** | **9.10** | **63.63** | **61.03** | **5.87** | **80.40** | **60.41** | **5.90** | **75.94** |

## 4.1 EXPERIMENTAL SETUP

**Metrics.** We evaluate performance using three standard metrics: Task Completion (**TC**), the success rate of stopping within 3m of the goal (↑); Shortest-path Distance (**SPD**), the final geodesic distance to the goal (↓); and normalized Dynamic Time Warping (**nDTW**), which measures path fidelity against the expert trajectory (↑). Formally, for $N$ episodes, given the final distance to goal $d_i$, expert path length $L_i$, and warping cost $\mathrm{DTW}_i$:

$$\mathrm{TC} = \frac{1}{N}\sum_{i=1}^{N}\mathbf{1}[d_i \leq 3], \quad \mathrm{SPD} = \frac{1}{N}\sum_{i=1}^{N}d_i, \quad \mathrm{nDTW} = \frac{1}{N}\sum_{i=1}^{N}\exp\bigl(-\mathrm{DTW}_i/L_i\bigr).$$

**Backbones and Implementation.** We integrate Mem4Nav into three diverse backbones to test its effectiveness: **(1) Hierarchical Modular Pipeline** is a fully modular, non–end-to-end system: a large language model generates scene descriptions, which are embedded and fed into our sparse octree + semantic graph builder; a hierarchical planner then decomposes the instruction into landmark, object and motion subgoals; and a lightweight policy network fuses planner outputs with retrieved memory to select actions. This pipeline was specifically devised by us to rigorously evaluate the performance of the memory module. **(2) VELMA** (Schumann et al., 2024) is an LLM-based agent that uses text prompts for action generation; and **(3) FLAME** (Xu et al., 2025) is an MLLM agent with strided cross-attention over visual tokens. All models are trained under a unified three-phase schedule on a single NVIDIA A100 GPU for fair comparison. Further implementation details for all backbones can be found in Appendix A.3.

## 4.2 MAIN RESULTS

As summarized in Table 1, Mem4Nav consistently improves performance across all three backbones on both the Touchdown and Map2Seq datasets.

**Hierarchical Modular Pipeline.** This backbone sees the most significant improvements, with Mem4Nav boosting Task Completion (TC) by **+13.25 points** and nDTW by **+12.96 points** on Touchdown Dev. These substantial gains underscore the critical need for an explicit, structured memory system in agents that lack strong innate memory capabilities.

**VELMA (LLM-based).** Even with a powerful LLM, adding structured memory yields substantial benefits. Mem4Nav improves TC by **+5.46 points** and nDTW by over **10 points** on Touchdown Dev, demonstrating that our explicit spatio-temporal memory helps the LLM ground instructions more effectively in complex environments.

**FLAME (MLLM-based).** Our system also enhances the state-of-the-art MLLM agent, boosting TC by **+8.82 points** and nDTW by **+9.09 points** on Touchdown Dev. This shows that while FLAME's attention mechanism provides some implicit memory, Mem4Nav's explicit and hierarchical representation further refines its long-range coherence and path alignment.

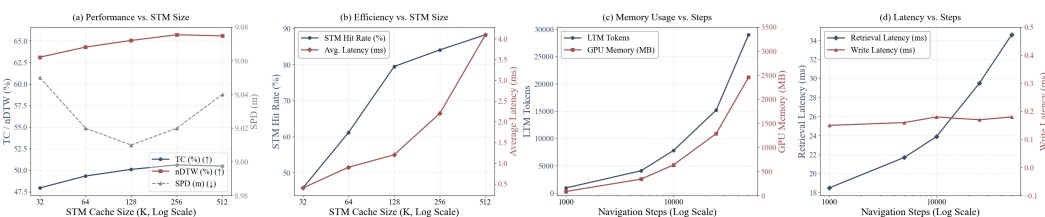

Figure 4: **Analysis of Mem4Nav's Hyper-parameters and Scalability. (a)** Navigation performance metrics (TC, nDTW, SPD) as a function of STM cache size. Performance gains saturate around K=128. **(b)** STM efficiency, showing that hit rate increases with cache size while latency also rises. **(c)** Long-horizon memory usage, illustrating the sub-linear growth of LTM tokens and GPU memory over 50,000 steps. **(d)** Long-horizon latency, demonstrating the near-constant O(1) write latency and the efficient O(log N) retrieval latency.

Overall, Mem4Nav consistently elevates navigation performance across diverse agent architectures. The pronounced improvements in Task Completion and nDTW confirm that our memory system successfully increases success rates and brings agent trajectories closer to expert demonstrations.

### 4.3 FURTHER ANALYSIS AND PARAMETER STUDIES

To further understand the behavior of Mem4Nav and validate its design choices, we conducted a series of analytical experiments focusing on hyper-parameter sensitivity and long-horizon scalability. We present the key findings in this section, with more detailed results available in the appendix.

The hyper-parameter and scalability analyses, visualized in Figure 4, offer deeper insights into Mem4Nav's design. As shown in plots (a) and (b), increasing the STM cache size ($K$) improves navigation performance (TC, nDTW) and hit rate, but these gains begin to saturate after $K = 128$. Given that latency continues to rise, this analysis validates our selection of $K = 128$ as a default configuration that strikes an optimal balance between high performance and low-latency local lookups. Furthermore, we assessed the system's long-horizon scalability. The results in plot (d) are particularly telling: the average octree write latency remains constant, empirically confirming the theoretical $O(1)$ average-time complexity of our hash-map-based sparse octree. Concurrently, the LTM retrieval latency scales sub-linearly, which is consistent with the efficient $O(\log N)$ query complexity of the HNSW index. Together, these findings demonstrate that Mem4Nav is not only effective but also highly efficient and scalable, making it well-suited for demanding, long-duration navigation tasks.

In addition to scalability, we analyzed the sensitivity to the STM replacement policy and robustness to depth estimation noise, summarized in Table 2. A balanced policy ($\lambda = 0.5$) that considers both recency and frequency is crucial for performance, outperforming policies that rely on only one factor. The system also exhibits graceful degradation under moderate depth sensor noise, though its dependency on input quality is clear. For a more comprehensive analysis, including the scalability of the semantic graph and a zero-shot transfer experiment showing Mem4Nav's generalization to indoor R2R environments, please refer to Appendix B.

Table 2: Analysis of STM Policy and Robustness to Depth Noise on Touchdown Dev.

| Category | Parameter / Condition | Key Finding & Impact on Task Completion (TC) |
|---|---|---|
| STM Policy | $\lambda = 0.0$ (Recency-only) | Vulnerable to forgetting key landmarks (TC: 48.91%). |
| | $\lambda = 0.5$ (Balanced, Default) | **Optimal trade-off**, achieving the highest performance (TC: 50.10%). |
| | $\lambda = 1.0$ (Frequency-only) | Fails to adapt to new context; Cognitive inertia (TC: 48.54%). |
| Robustness | Gaussian Depth Noise ($\sigma = 0.5$m) | Performance degrades gracefully (TC drops by 4.08 pp). |
| | 20% Depth Dropout | Highlights dependency on depth quality (TC drops by 5.54 pp). |

### 4.4 ABLATION STUDIES

We conduct a component-wise ablation study to validate the contribution of each module in Mem4Nav: the sparse octree, semantic graph, Long-Term Memory (LTM), and Short-Term Memory (STM).

Table 3: Component-wise ablations of Mem4Nav on Touchdown and Map2Seq. "w/o X" denotes removing component X from full Mem4Nav framework.

| Model | Touchdown Dev | | | Touchdown Test | | | Map2Seq Dev | | | Map2Seq Test | | |
|---|---|---|---|---|---|---|---|---|---|---|---|---|
| | TC↑ | SPD↓ | nDTW↑ | TC↑ | SPD↓ | nDTW↑ | TC↑ | SPD↓ | nDTW↑ | TC↑ | SPD↓ | nDTW↑ |
| **FLAME + full Mem4Nav** | **50.10** | 9.01 | **65.05** | **48.48** | 9.10 | **63.63** | **61.03** | 5.87 | **80.40** | **60.41** | 5.90 | **75.94** |
| FLAME +Mem4Nav w/o Octree | 48.72 | 9.08 | 60.90 | 47.52 | 9.18 | 58.85 | 59.10 | 5.95 | 76.20 | 58.35 | 6.02 | 73.10 |
| FLAME +Mem4Nav w/o Semantic Graph | 44.40 | 9.25 | 62.10 | 45.83 | 9.55 | 61.42 | 58.50 | 6.10 | 78.00 | 56.90 | 6.20 | 74.50 |
| FLAME +Mem4Nav w/o LTM | 47.28 | 9.03 | 64.02 | 47.90 | 9.12 | 62.70 | 60.10 | 5.88 | 79.20 | 59.00 | 5.95 | 75.50 |
| FLAME +Mem4Nav w/o STM | 48.67 | **9.00** | 62.35 | 48.10 | **9.08** | 62.10 | 60.50 | **5.87** | 79.80 | 59.90 | **5.90** | 75.30 |
| **VELMA + full Mem4Nav** | **35.29** | **12.16** | **55.35** | **34.04** | 12.90 | **48.82** | **58.33** | 6.01 | **75.06** | **56.84** | 6.10 | **72.71** |
| VELMA +Mem4Nav w/o Octree | 34.05 | 12.50 | 53.00 | 32.80 | 13.20 | 45.90 | 57.00 | 6.20 | 73.21 | 55.00 | 6.30 | 70.50 |
| VELMA +Mem4Nav w/o Semantic Graph | 33.50 | 12.70 | 51.50 | 32.20 | 13.40 | 44.21 | 56.20 | 6.33 | 71.20 | 54.30 | 6.45 | 69.10 |
| VELMA +Mem4Nav w/o LTM | 31.32 | 13.20 | 47.01 | 29.85 | 14.06 | 41.40 | 54.00 | 7.12 | 67.00 | 51.50 | 7.10 | 62.50 |
| VELMA +Mem4Nav w/o STM | 33.14 | **12.16** | 49.50 | 32.50 | 12.91 | 47.05 | 56.55 | 6.02 | 74.00 | 55.50 | **6.10** | 71.00 |
| **Hierarchical + full Mem4Nav** | **45.18** | **11.21** | **59.03** | **42.21** | **11.95** | **56.36** | **58.19** | 5.49 | **74.74** | **57.64** | 5.54 | **73.57** |
| Hierarchical +Mem4Nav w/o Octree | 39.31 | 12.50 | 52.42 | 38.25 | 12.91 | 50.45 | 55.85 | 6.04 | 70.32 | 55.20 | 5.82 | 67.35 |
| Hierarchical +Mem4Nav w/o Semantic Graph | 35.56 | 12.24 | 52.35 | 34.05 | 12.76 | 46.04 | 54.46 | 6.05 | 71.15 | 52.52 | 6.13 | 69.20 |
| Hierarchical +Mem4Nav w/o LTM | 33.42 | 12.54 | 51.23 | 31.52 | 12.73 | 47.30 | 55.42 | 6.13 | 72.02 | 52.34 | 6.02 | 66.23 |
| Hierarchical +Mem4Nav w/o STM | 41.34 | 11.25 | 53.31 | 38.50 | 11.98 | 52.00 | 56.00 | 5.51 | 69.85 | 53.50 | 5.57 | 67.26 |

Each component is removed individually to assess its impact on the three backbones, with results presented in Table 3.

**Hierarchical Modular Pipeline.** This agent is highly sensitive to all components. Removing the LTM causes a catastrophic drop in task success (TC falls by 11.76 points), demonstrating its critical role in long-term recall. Similarly, ablating the semantic graph severely hampers high-level planning, resulting in a 9.62-point drop in TC.

**VELMA (LLM-based).** The LLM agent relies heavily on explicit memory. Ablating the STM significantly degrades path fidelity (nDTW drops by 5.85 points), while removing the LTM harms task completion (TC drops by 3.97 points). The impact of removing spatial modules is less severe, as the LLM can partially compensate through its internal reasoning.

**FLAME (MLLM-based).** Despite its implicit memory from cross-attention, the MLLM still requires explicit spatial structures. Removing the semantic graph significantly reduces task completion (TC drops by 5.70 points), while ablating the sparse octree hurts path fidelity (nDTW drops by 4.15 points), proving that both high-level and fine-grained spatial awareness remain critical. Overall, the ablations confirm that while LLM/MLLM agents can partially compensate for missing components, they still derive distinct benefits from Mem4Nav's explicit memory and hierarchical spatial representations.

## 5 CONCLUSION AND DISCUSSION

In this work, we introduced **Mem4Nav**, a hierarchical long-short memory system designed to enhance VLN agents in complex urban environments. By integrating a dual-component spatial map with a reversible Transformer-based memory, Mem4Nav achieves both efficient, lossless long-term recall and rapid local adaptation. Our experiments empirically demonstrate that Mem4Nav substantially improves navigation success and path fidelity across diverse backbones, from modular pipelines to state-of-the-art LLM and MLLM agents. Ablation studies further confirm that each component of our hierarchical memory architecture is critical to its overall effectiveness.

Furthermore, our work engages with the critical debate on explicit versus implicit knowledge for embodied AI. The performance gains on powerful MLLM backbones highlight the value of a hybrid approach, where a structured external memory augments a foundation model's generalist reasoning. This aligns with concurrent research demonstrating that explicitly integrating 3D semantic maps is a highly effective strategy for instruction-guided navigation with large models (Wang et al., 2025). We view Mem4Nav as a step towards building dynamic, grounded world models for agents. As recent surveys on spatial intelligence emphasize, the key future challenge is to evolve such memory systems from single-mission data logs into true lifelong learning frameworks that can adapt to changing environments (Feng et al., 2025). This advancement would require new mechanisms for memory consolidation and selective forgetting, ultimately enabling agents to move beyond simple trajectory following to perform complex, causal reasoning about their actions in a continuously changing world.

ETHICS STATEMENT

The work presented in this paper is methodological in nature, focusing on the development of Vision-Language Navigation. To the best of our knowledge, our proposed methods do not introduce any new ethical concerns.

REPRODUCIBILITY STATEMENT

To facilitate the verification of our results, the implementation code for our algorithm and the main baselines is provided in the anonymous code link and the appendix.

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

## USE OF LARGE LANGUAGE MODELS

We utilized a large language model to enhance the language and clarity of our manuscript. Specifically, we employed Gemini 2.5 flash with the following prompt to refine the initial draft: *I am writing an academic paper in English. Please polish the following draft so that it adheres to the conventions of academic writing.*

## A  APPENDIX

### A.1  USE OF LARGE LANGUAGE MODELS

We utilized a large language model to enhance the language and clarity of our manuscript. Specifically, we employed Gemini 2.5 flash with the following prompt to refine the initial draft: *I am writing an academic paper in English. Please polish the following draft so that it adheres to the conventions of academic writing.*

### A.2  ALGORITHM IN DETAIL

#### A.2.1  SPARSE OCTREE LEAF INSERTION AND UPDATE

We discretize 3D world coordinates into a hierarchical octree of maximum depth $\Lambda$. Each node at level $\ell \in \{0, \dots, \Lambda\}$ represents an axis–aligned cube of side length $L/2^{\ell}$. Only leaves that have been visited or contain relevant observations are instantiated and stored in a hash table for $O(1)$ lookup.

**Morton Key Computation.** Given a continuous agent position $p_t = (x_t, y_t, z_t)$, we quantize to integer coordinates $\bar{p}_t = (\lfloor x_t 2^{\Lambda}/L \rfloor, \dots) \in \{0, \dots, 2^{\Lambda} - 1\}^3$, then interleave bits to form a Morton code (Z-order curve)

$$\kappa(p_t) \;=\; \mathrm{InterleaveBits}(\bar{x}_t, \bar{y}_t, \bar{z}_t) \;\in\; \{0, \dots, 2^{3\Lambda} - 1\}.$$

This single integer $\kappa$ uniquely identifies the leaf voxel at depth $\Lambda$.

**Octree Leaf Update.** On each visit to $p_t$:

- Compute leaf key $\kappa_t$.
- If $\kappa_t$ not in hash table $\mathcal{H}$, create new leaf entry $\mathcal{O}_{\kappa_t} = \{\theta^r_{\kappa_t}, \theta^w_{\kappa_t}, B_{\kappa_t}\}$, where $B_{\kappa_t}$ stores the cube bounds.
- Retrieve $\theta^r \leftarrow \theta^r_{\kappa_t}$.
- Fuse current embedding $v_t$ into memory via reversible update:

$$\theta^w_{\kappa_t} \;\leftarrow\; \mathcal{R}\big(\theta^r; v_t\big), \qquad \theta^r_{\kappa_t} \;\leftarrow\; \theta^w_{\kappa_t}.$$

All operations (hash lookup, token update) cost $O(1)$ average time; the Morton code computation and bit interleaving cost $O(\Lambda)$.

**Monocular Depth Estimation with UniDepth**

To recover metric depth from single RGB panoramas, we adopt UniDepth, a universal monocular metric depth estimator that directly predicts dense 3D points without requiring known camera intrinsics at test time. UniDepth incorporates a self-promptable camera module that outputs a dense spherical embedding of azimuth and elevation angles, which conditions a depth module via cross-attention, and uses a pseudo-spherical ($\theta$, $\phi$, zlog) output representation to disentangle camera pose from depth prediction :contentReference[oaicite:0]index=0. A geometric-invariance loss further enforces consistency between depth features under different geometric augmentations :contentReference[oaicite:1]index=1:contentReference[oaicite:2]index=2.

**Integration into Mem4Nav.** At each time step $t$, given the current panorama $I_t$, we run UniDepth to obtain a dense depth map $D_t$ and the camera embedding $C_t$. We then unproject each pixel $(u, v, id)$ via the predicted pseudo-spherical outputs to form a local point cloud

$$P_t = \big\{(x_i, y_i, z_i) \;\big|\; (u_i, v_i, z_i) \in D_t\big\},$$

which supplies the $z$-coordinate for Morton code quantization in the sparse octree. We augment the visual feature vector $v_t^{\text{RGB}}$ (from the perception backbone) with a depth feature vector $v_t^{\text{Depth}} = \text{MLP}\big(\text{CA}(F_t, C_t)\big)$—the cross-attention output of the depth module—to form the fused embedding

$$v_t = \big[v_t^{\text{RGB}}; v_t^{\text{Depth}}\big].$$

This fused embedding is then written into both (i) the octree leaf at key $\kappa(p_t)$ and (ii) any semantic graph node created or updated at position $p_t$, via the reversible token write operator . By integrating metric depth in this way, Mem4Nav's hierarchical spatial structures gain true 3D scale awareness, improving both the precision of voxel indexing and the semantic graph's landmark localization.

[H] [1] **Input:** position $p_t$, embedding $v_t$, hash table $\mathcal{H}$ $\bar{p} \leftarrow \text{Quantize}(p_t)$ $\kappa \leftarrow \text{InterleaveBits}(\bar{p})$ $\kappa \notin \mathcal{H}$ initialize $\theta^r, \theta^w \sim \mathcal{N}(0, I_d)$ $\mathcal{H}[\kappa] \leftarrow (\theta^r, \theta^w, B_\kappa)$ $(\theta^r, \theta^w, B) \leftarrow \mathcal{H}[\kappa]$ $\theta^w \leftarrow \mathcal{R}(\theta^r; v_t)$ Reversible write $\theta^r \leftarrow \theta^w$ $\mathcal{H}[\kappa].\theta^r \leftarrow \theta^r$, $\mathcal{H}[\kappa].\theta^w \leftarrow \theta^w$

### A.2.2 SEMANTIC NODE & EDGE UPDATE

While the octree captures raw geometry, many navigation cues come from salient landmarks or decision points (e.g. intersections, points of interest). We maintain a dynamic graph $\mathcal{G} = (\mathcal{V}, \mathcal{E})$ whose nodes $u \in \mathcal{V}$ correspond to important locations and whose edges $(u_i, u_j) \in \mathcal{E}$ record traversability and cost.

**Node Creation and Token Fusion.** Whenever the agent's VLM detects a trigger phrase (e.g. turn left at the statue) or a high semantic change in embedding:

$$\exists\, u' \in \mathcal{V} : \ \|v_t - \phi(u')\| \le \delta$$

where $\phi(u)$ is the aggregate descriptor of node $u$. If no existing node is within threshold $\delta$, we create a new node:

$$u_{\text{new}}.\ p \leftarrow p_t, \quad (\theta_u^r, \theta_u^w) \sim \mathcal{N}(0, I_d),$$

and add $u_{\text{new}}$ to $\mathcal{V}$. Then we fuse the embedding:

$$\theta_u^w \ \leftarrow\ \mathcal{R}(\theta_u^r; v_t), \qquad \theta_u^r \ \leftarrow\ \theta_u^w.$$

**Edge Addition and Weighting.** Each time the agent moves from node $u_{t-1}$ to $u_t$, we add or update edge $(u_{t-1}, u_t)$ with weight

$$w_{t-1,t} \ =\ \alpha\, \|p_{t-1} - p_t\|_2 \ +\ \beta\, c_{\text{instr}},$$

where $\alpha, \beta$ balance Euclidean distance and instruction cost $c_{\text{instr}}$ (e.g. number of turns). If the edge already exists, we average weights to smooth noise.

[H] [1] **Input:** embedding $v_t$, position $p_t$, graph $\mathcal{G}$ found $\leftarrow \arg\min_{u \in \mathcal{V}} \|v_t - \phi(u)\|$ $\|v_t - \phi(\text{found})\| > \delta$ create new node $u$ with $u.p \leftarrow p_t$, random tokens $\mathcal{V} \cup \{u\}$ $u \leftarrow \text{found}$ $(\theta_u^r, \theta_u^w) \leftarrow u.\text{tokens}$ $\theta_u^w \leftarrow \mathcal{R}(\theta_u^r; v_t)$, $\theta_u^r \leftarrow \theta_u^w$ $u.\text{tokens} \leftarrow (\theta_u^r, \theta_u^w)$ previous node $u_{\text{prev}}$ exists compute $w \leftarrow \alpha \|p_t - u_{\text{prev}}.p\| + \beta\, c_{\text{instr}}$ add/update edge $(u_{\text{prev}}, u)$ with weight $w$

### A.2.3 LONG–TERM MEMORY WRITE AND RETRIEVAL

The long-term memory module stores and retrieves spatially anchored observations in a lossless, compressed form. When writing, each spatial element's existing memory tokens are updated by fusing in the new observation embedding via a reversible transform, replacing the old token. For retrieval, the current observation and position are projected into a query vector, which is used to perform an approximate nearest-neighbor search over all stored tokens. The top matches are then inverted through the reversible transform to reconstruct their original embeddings and associated spatial information, which are returned for downstream reasoning.

[H] [1] LTM_Writeelement $s$, embedding $v_t$ $(\theta^r, \theta^w) \leftarrow \text{Tokens}(s)$ $\theta^w \leftarrow \mathcal{R}(\theta^r \| v_t)$ $\theta^r \leftarrow \theta^w$ Tokens$(s) \leftarrow (\theta^r, \theta^w)$ LTM_Retrievequery $(v_t, p_t)$ $q \leftarrow \text{Proj}([v_t; p_t])$ $\{s_i\} \leftarrow \text{HNSW\_NN}(q)$ each $s_i$ $\widehat{v}_i \leftarrow \mathcal{R}^{-1}(\theta_{s_i}^r)$ $\widehat{p}_i \leftarrow \pi_p(\widehat{v}_i)$, $\widehat{d}_i \leftarrow \pi_d(\widehat{v}_i)$ $\{(\widehat{p}_i, \widehat{d}_i)\}$

**HNSW Index Configuration and Usage**

We use the **Hierarchical Navigable Small World** (HNSW) algorithm to index and query our collection of read-tokens $\{\theta_s^r\} \in \mathbb{R}^d$. HNSW organizes vectors into a multi-layer graph where each layer is a **small-world proximity** graph, enabling logarithmic-scale search complexity and high recall in practice.

**Index Construction.** HNSW incrementally inserts tokens one by one. Each new token $\theta^r$ is assigned a maximum layer $L$ drawn from a geometric distribution (probability $p = 1/M$), so higher layers are sparser. For each layer $\ell \leq L$:

- Starting from an entry point at the topmost nonempty level, perform a **greedy** search: move to the neighbor closest (by cosine distance) to $\theta^r$ until no closer neighbor is found.
- Maintain a candidate list of size `efConstruction` to explore additional connections beyond the greedy path.
- Select up to `M` closest neighbors from the candidate list and bidirectionally link them with $\theta^r$.

This builds a nested hierarchy of proximity graphs: the top layer provides long-range jumps, while lower layers refine locality.

**Querying (Search).** To find the $k$ nearest tokens to a query $q$:

- **Entry-point jump:** Begin at the top layer's entry point; greedily traverse neighbors to approach $q$.
- **Layer descent:** At each lower layer, use the best candidate from the previous layer as the starting point, repeating the greedy step.
- **Beam search at base layer:** At layer 0, perform a best–first search with a dynamic queue of size `efSearch`. Expand the closest candidate by examining its neighbors, inserting unseen neighbors into the queue, and retaining the top `efSearch` candidates.
- **Result selection:** Once no closer candidates remain or budget is exhausted, output the top $k$ tokens from the queue.

**Hyperparameters and Complexity.**

- M: maximum number of neighbors per node (e.g. 64).
- efConstruction: candidate list size during insertion (e.g. 500), trading off build time vs. graph quality.
- efSearch: candidate list size during queries (e.g. 200), controlling recall vs. search latency.

### A.2.4 SHORT–TERM MEMORY INSERT & RETRIEVE

The short-term memory module maintains a compact, fixed-size buffer of the most recent observations relative to the agent's current position. Whenever the agent perceives a new object, the module computes its position with respect to the current node and checks if an entry for that object already exists. If it does, the entry is refreshed with the latest embedding and timestamp and its access count is increased. If the object is new and there is still room in the buffer, a new entry is appended. Once the buffer is full, the least valuable entry—determined by a balance of how often and how recently it has been used—is removed to make space for the new observation. When the agent needs to recall local context, the module filters entries within a small spatial neighborhood of the agent's position and returns those whose stored embeddings best match the current observation. This mechanism ensures fast, spatially anchored retrieval without unbounded memory growth.

**Insertion and Update.**

- Compute relative position $p_{\text{rel}}$.
- If an entry with same object $o$ exists, update its $v, \tau$, and increment freq.
- Else if $|\text{STM}| < K$, append new entry with freq $= 1$.
- Otherwise, evict $e_{\text{evict}}$ and insert new entry.

[H] [1] STM_Insert$(o, p_t, v_t)$ $p_{\text{rel}} \leftarrow p_t - p_{u_c}$ exists $e_i.o = o$ $e_i.v \leftarrow v_t$, $e_i.\tau \leftarrow t$, $e_i.\text{freq}+ = 1$ $|\text{STM}| < K$ append $e = (o, p_{\text{rel}}, v_t, t, \text{freq} = 1)$ evict $\arg\min_i \left[\lambda \text{freq}_i - (1 - \lambda)(t - \tau_i)\right]$ insert

new $e$ STM_Retrieve$v_t, p_t$ $q_{\text{rel}} \leftarrow p_t - p_{u_c}$ $\mathcal{C} \leftarrow \{e_i : \|e_i.p_{\text{rel}} - q_{\text{rel}}\| \leq \epsilon\}$ compute $s_i = \cos(v_t, v_i)$ for $e_i \in \mathcal{C}$ top–$k$ entries by $s_i$

## A.3 MORE DETAILS ON IMPLEMENTATION AND BACKBONES

This appendix provides the full implementation details for all three backbone agents and the shared training regimen. Readers are referred to the main text (Section 3.2) for a concise summary; here we enumerate every architectural choice, hyperparameter, and integration point.

### A.3.1 HIERARCHICAL MODULAR PIPELINE

**Open-Vocabulary Perception Module** We preprocess each panoramic observation by extracting five overlapping $90°$ crops at headings spaced by $45°$. Each crop is passed through GPT-4V to generate a free-form scene description, then through GroundingDINO (confidence threshold 0.4) and Segment Anything to obtain open-vocabulary object detections with fine-grained masks. Simultaneously, the RGB–D image (512×512, $90°$ FOV) is projected into a local point cloud using known camera intrinsics and the agent's pose. The resulting captions, object labels, and local 3D points are concatenated into semantic vectors (512 d) that serve as the perception output.

**Hierarchical Semantic Planning** From the perception vectors, we prompt GPT-4V with a structured JSON template to extract an ordered list of landmarks mentioned in the instruction. For each landmark, we group the relevant 3D points and object detections to form a semantic region proposal. Once the landmark sequence is obtained, we decompose each segment into a series of waypoint goals: first selecting the nearest graph node or region centroid, then planning a grid-based path using A on a 0.5 m resolution lattice and motion primitives of forward or $\pm15°$ turns. This three-tiered planning—landmark ordering, region centroids, and primitive-level path—ensures both high-level coherence and low-level feasibility in outdoor environments.

**Reasoning and Decision Integration** At each step, the current perception embedding, the next waypoint from the semantic planner, and any retrieved memory summaries are combined into a single context vector. We first attempt to retrieve from the short-term cache (capacity 128, FLFU policy with equal weight on frequency and recency, spatial radius 3 m); on cache miss we fall back to an HNSW-based long-term lookup (index size 10 K, retrieve top 3). Retrieved summaries are rendered as concise natural-language bullets under Past memory: in the GPT-4V prompt. The final prompt (capped at 512 tokens) is fed into GPT-4V with a greedy decoding setting (temperature 0.0) and a constrained vocabulary mask allowing only $\{$forward,left,right,stop$\}$. This unified prompt-based decision ensures that modular perception, planning, and memory seamlessly inform each navigation action.

**Integration of Mem4Nav** At each time step, the current visual embedding is written into both the sparse octree and the semantic topology graph via reversible memory tokens, and the same embedding is inserted into the short-term cache (evicting entries according to the replacement policy when full). When the hierarchical planner emits the next waypoint, we first query the STM for any recent observations . If fewer than two relevant entries are found, we fall back to an HNSW-based LTM lookup over all read-tokens in the octree and graph (index size 10 K), decode the selected tokens back into spatial and descriptor information, and render them as concise Past memory: bullets. These memory summaries, together with the next waypoint and the perception output, are concatenated into the GPT-4V prompt (capped at 512 tokens) before decoding. By injecting both fine-grained local context and lossless long-range recalls into the decision prompt—while still respecting our constrained action vocabulary—Mem4Nav seamlessly augments the modular pipeline with structured, multi-scale memory.

### A.3.2 VELMA BACKBONE (DETAILED)

In this section we provide the full implementation details for the VELMA backbone used in our experiments, so that readers can exactly reproduce the behavior and performance reported in the main text.

**Model Checkpoint and Dependencies** We use the CLIP-ViT/L14 vision encoder and the LLaMA-7B language model decoder. All weights are frozen except where noted below.

Figure 5: Overview of VELMA

**Visual Preprocessing**

- **Input panoramas:** we sample four 90°-FOV crops from each 360° panorama at headings $\{0, 90, 180, 270\}$.
- **Resize & normalize:** each crop is resized to $224 \times 224$ pixels, normalized with ImageNet mean/std.
- **Patch tokenization:** the CLIP-ViT/L14 splits the $224 \times 224$ input into $14 \times 14$ patches (total 196 tokens), each mapped to a 768-dim embedding.

**Memory Integration**

- **STM retrieval:** we compute cosine similarity between the current CLIP-ViT embedding for each detected object and each STM entry's 256-dim vector. We select up to $k = 4$ entries with similarity $> 0.5$.
- **LTM retrieval:** if fewer than 2 STM entries pass the threshold, we query the HNSW index built over all read-tokens ($d = 256$, M=16, ef=200), retrieve the top 3, and run the reversible Transformer inverse to decode their stored embeddings back into ($p$, desc, state) triples.
- **Natural-language summarization:** each retrieved entry is rendered as a one-line bullet under Past memory: using the template:

$$\text{at } (x_j, y_j)\text{: saw } o_j, \text{ status } s_j.$$

with $x_j, y_j$ rounded to one decimal.

**Decoding and Action Selection**

- The full prompt (up to 512 tokens) is fed into the LLaMA-7B decoder with a temperature of 0.0 (greedy).
- We apply a constrained vocabulary mask so that only the four actions $\{\texttt{forward}, \texttt{left}, \texttt{right}, \texttt{stop}\}$ can be generated.
- The single generated token is mapped directly to the discrete action.

### A.3.3 FLAME BACKBONE (DETAILED)

The FLAME backbone is built upon the Otter architecture (CLIP-ResNet50 encoder + LLaMA-7B decoder) with strided cross-attention. Below we describe every component and training detail necessary for exact reproduction.

#### MODEL ARCHITECTURE

The vision frontend is a CLIP ResNet-50 network, taking each pano crop of size $224 \times 224$ and extracting a $7 \times 7 \times 2048$ feature map. A linear projection reduces each spatial vector to 512 d:

$$f_{t,i} = W_{\text{proj}}\big(\text{ResNet50}(I_{t,i}^{\text{rgb}})\big) + b_{\text{proj}}, \quad i = 1, \dots, 49.$$

These 49 patch embeddings $f_{t,1:49} \in \mathbb{R}^{512}$ form the visual token sequence $O_t$.

The language backbone is a 7 B LLaMA model (32 layers, $d_{\text{model}} = 4096$, 32 heads). We interleave four cross-attention modules into layers 8, 12, 16, and 20. Each cross-attention takes as queries the LLaMA hidden states $h_\ell \in \mathbb{R}^d$ and as keys/values the concatenation of:

$$\left[ O_{t-K+1}, O_{t-K+1+2}, \ldots, O_t \right] \in \mathbb{R}^{K \times 512},$$

with $K = 5$ and temporal stride 2. This strided attention allows the model to attend past panoramas at intervals, reducing quadratic cost while preserving longer-range context.

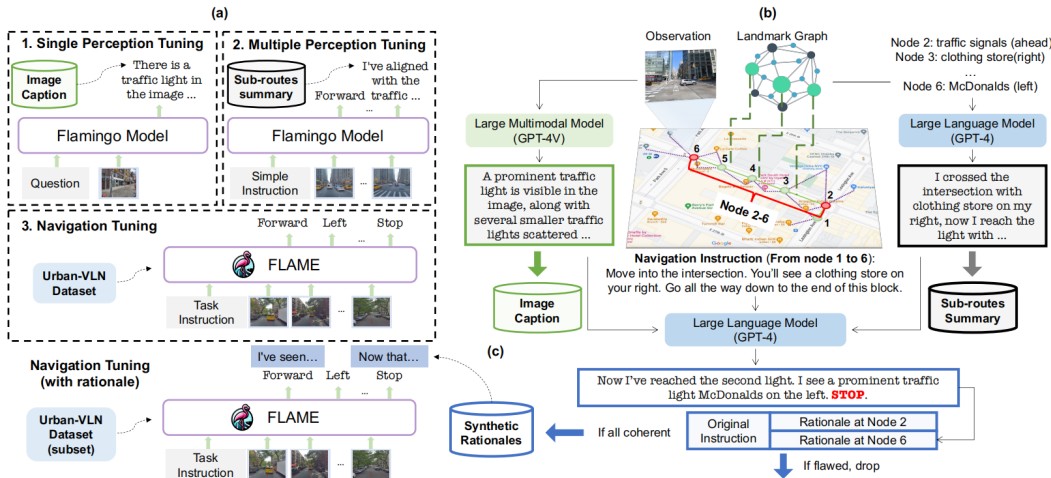

Figure 6: Overview of FLAME

## MEMORY INTEGRATION

After obtaining the visual token sequence $O_t$, we perform memory retrieval:

- **Long-Term Memory (LTM):** Query with the current merged embedding $q_t = \text{MLP}([h_{t-1}; \overline{f}_t])$ against the HNSW index of all stored read-tokens $\{\theta_j^r\}$. Retrieve the top $m = 3$ tokens $\theta_{j_1}^r, \theta_{j_2}^r, \theta_{j_3}^r \in \mathbb{R}^{256}$.
- **Short-Term Memory (STM):** Filter cache entries by relative coordinate proximity $\|p_t - p_{u_j}\| < \epsilon$, compute cosine similarity with $q_t$, and select the top $k = 2$ vectors $s_{t,1}, s_{t,2} \in \mathbb{R}^{256}$.

We then augment the cross-attention key/value inputs by concatenating these memory vectors along the spatial axis:

$$\text{KV}_t = \left[ f_{t,1:49}; \theta_{j_{1:3}}^r; s_{t,1:2} \right] \in \mathbb{R}^{(49+5) \times 512},$$

with learnable linear mappings applied to project $\theta^r$ and $s$ into 512 d.

### A.4 FAILURE CASES

Despite the substantial gains of Mem4Nav, our Touchdown and Map2Seq evaluations reveal three dominant failure modes:

**Depth-induced mapping errors.** Monocular depth estimates from UniDepth can be highly inaccurate in low-texture regions (e.g. blank building façades) or under extreme lighting, causing voxels in the sparse octree to be misplaced by several meters. These misregistrations propagate into both LTM writes and spatial lookups, leading the agent to misjudge its surroundings and execute incorrect turn or stop actions.

- **Scenario:** In the panorama shown in the case, the agent faces a long stretch of repetitive, uniform window façades with minimal texture.
- **Voxel Misregistration:** This bias shifts the corresponding octree leaves by 2–3 voxels (leaf size = 1 m), causing building-edge voxels to be placed several meters into the adjacent roadway.

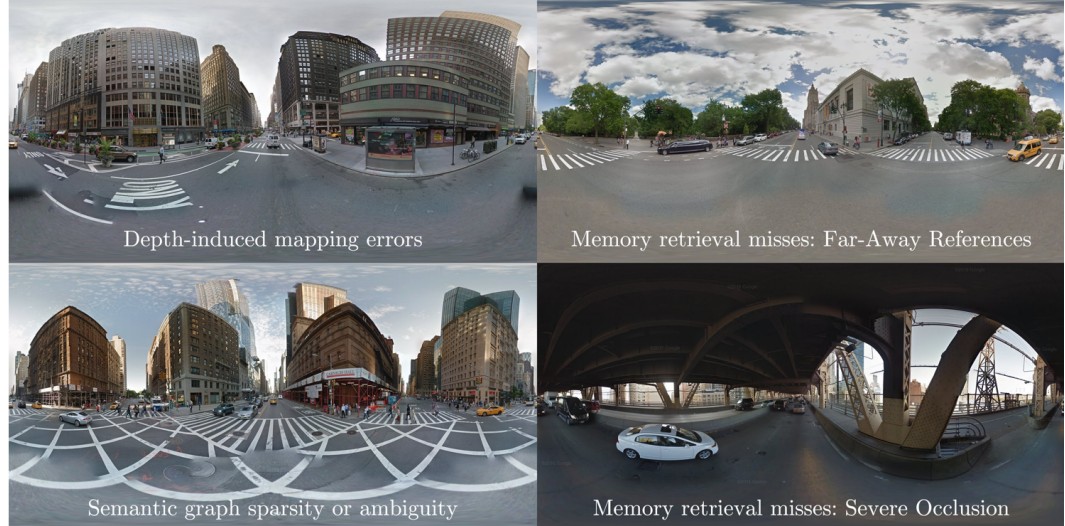

Figure 7: **Representative Failure Cases of Mem4Nav.** Despite substantial overall gains, we identify four dominant error modes: (a) *Depth-induced mapping errors* arise when monocular depth estimates misplace voxels on low-texture façades, corrupting both octree writes and lookups; (b) *Memory retrieval misses for far-away references* occur because distant landmarks lie outside instantiated spatial bins and yield no sufficiently similar tokens; (c) *Semantic graph sparsity or ambiguity* results when subtle or partially occluded landmarks (e.g. crosswalk markings) fail node creation, breaking planned routes; and (d) *Memory retrieval misses under severe occlusion* happen when landmarks hidden by overhead structures cannot be matched by either STM filtering or LTM HNSW search.

- **Graph and Memory Impact:**
  - The **semantic graph** creates the next intersection node 4 m too far ahead, so the agent believes it must walk past the actual corner.
  - The **STM cache** retrieves recent building edge observations at the wrong relative coordinates, confusing the local planner.
  - Long-term recall of the corner store landmark is falsely triggered before the real intersection, leading to a premature turn.

**Semantic graph sparsity or ambiguity.** Our threshold-based node creation occasionally fails to instantiate graph nodes for subtle or partially occluded landmarks (e.g. crosswalk markings, small storefront signs). When a required intersection node is missing, the planner cannot recover the intended route sequence, resulting in the agent overshooting turns or taking suboptimal detours.

- **Scenario:** In the panorama of a complex intersection with multiple crosswalk markings , the agent's landmark detector labels each zebra-stripe segment as a distinct crosswalk object.

- **Graph Node Explosion:** Our descriptor-distance threshold $\delta$ causes each segmented stripe to spawn a separate node, resulting in 12 crosswalk nodes clustered around the same intersection rather than a single intersection node.

- **Missing Intersection Node:** Because no single node accumulates enough repeated visits (all crosswalk nodes receive only one write), the true intersection landmark is never consolidated, leaving a gap in the semantic graph at that decision point.

- **Routing Consequence:**
  - The global planner fails to include the intended turn at crosswalk step, treating the next valid node as two blocks ahead.
  - The local planner, flooded with near-duplicate crosswalk STM entries at slightly different offsets, cannot decide when to pivot, causing the agent to overshoot the turn by an average of 5.2 m.

**Memory retrieval misses.** The STM cache sometimes fails to match recently observed landmarks when the agent's viewpoint shifts rapidly. Likewise, under heavy index loads, the HNSW ANN search can return suboptimal long-term tokens, causing the policy to fall back on stale or irrelevant memories.

Landmarks partially or fully blocked by passing vehicles, pedestrian crowds, or temporary structures (e.g. scaffolding) reduce feature visibility, causing both STM spatial filters and LTM similarity search to miss the stored tokens. For instance, the target landmark (archway under the bridge) is largely hidden by the overhead girders. The visual detector only extracts low-contrast fragments, producing an embedding that differs significantly from the original octree and graph tokens. During STM spatial filtering the relative positions match, but the cosine similarity falls below threshold. Likewise, the HNSW search in LTM does not return the hidden archway token. Consequently, the agent cannot recall the landmark and incorrectly continues past the underpass, deviating from the instructed path. Furthermore, instructions that refer to distant landmarks beyond the STM radius and whose tokens in LTM are too sparsely distributed in the octree or graph layers, so even HNSW search returns no sufficiently close vectors. Both issues lead to degraded local decisions and trajectory drift.

## A.5 REAL-WORLD DEPLOYMENT

**Deployment Setup.** We ported Mem4Nav onto a robotic dog under ROS Melodic with a RGB camera. The onboard RGB camera captures $125°$ field-of- view images at 10 Hz, which are processed by UniDepth for per-pixel monocular depth estimation. For trialing, we **manually designed** the following six-step navigation protocol through a mixed urban block:

1. Proceed eastbound through the cross-type intersection.

2. Maintain eastbound traversal at the T-junction adjacent to the grasslands.

3. Execute a left turn (southward) at the T-junction located at the northeastern vertex of the Sports Instructors Training Base.

4. Just before the next intersection, observe a blue bike on the right in front of a stadium.

5. Initiate a left turn (southward) at the T-junction at the northwestern quadrant of a brown building, where a tall man is leaning on a tree.

6. Terminate navigation at the designated coordinates: a playground with an orange safety light.

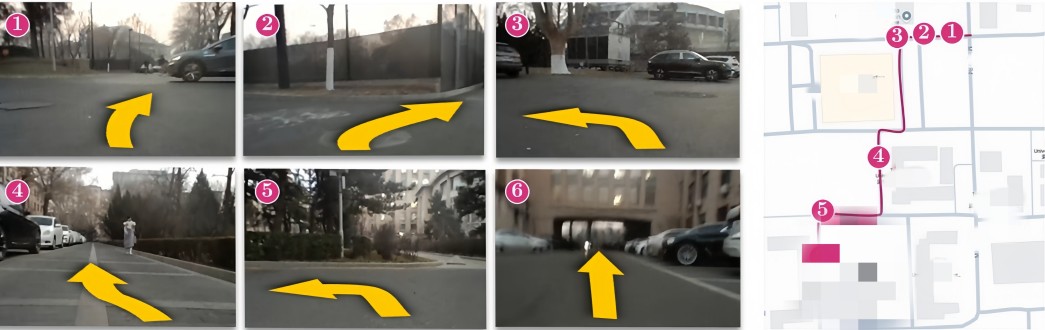

Figure 8: Route of a campus real world trial of MemNav.

**Experimental Results.** We conducted 30 real-world trials across varying times of day and pedestrian densities. Mem4Nav achieved a success rate of 70% (21/30 runs) defined by stopping within 3 m of the goal.

**Failure Cases.** Among the nine failures, two predominant modes emerged: (1) *Depth-induced mapping drift*: uniform asphalt and large blank façades caused UniDepth errors, leading to misregistered octree voxels and missed turn decisions; (2) *Dynamic occlusions*: clusters of pedestrians and parked vehicles intermittently blocked key landmarks, resulting in STM cache misses and incorrect semantic graph traversals. These highlight the need for robust depth correction and dynamic-object filtering in future real-world deployments.

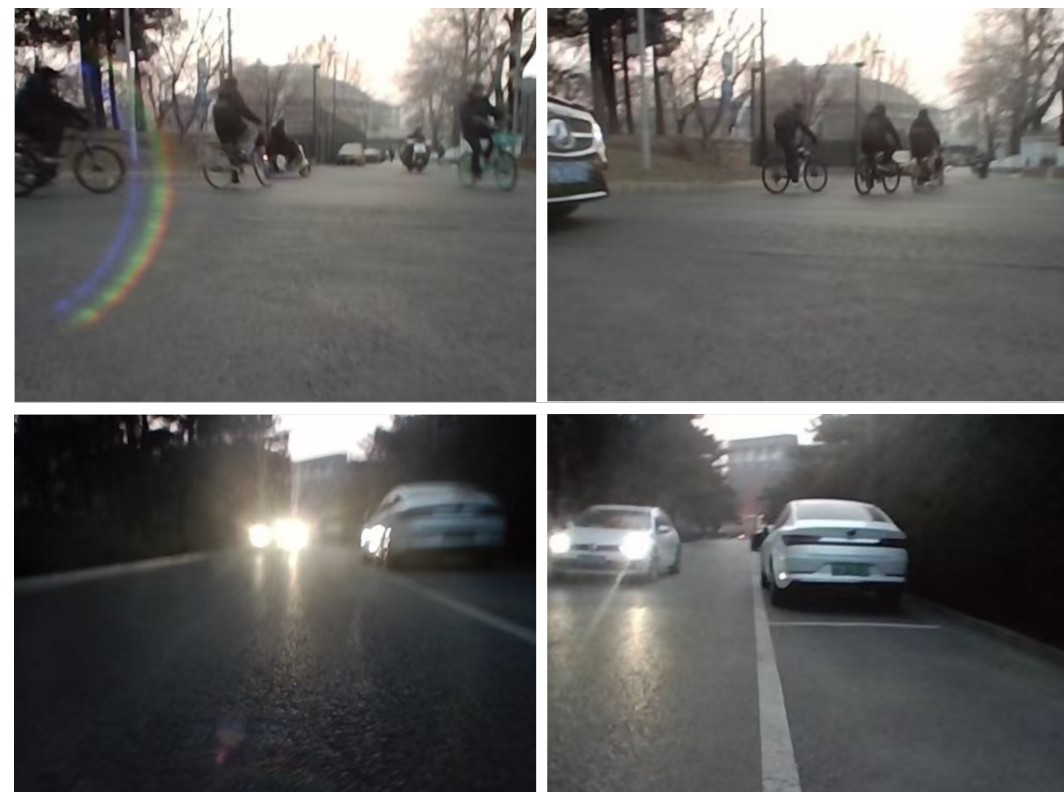

Figure 9: Failure Cases. (a) The agent came to a halt at a busy uncontrolled intersection, where the substantial volume of vehicular and pedestrian traffic rendered it incapable of determining an opportune moment to proceed. (b) The lights from high-velocity oncoming vehicles compromised the agent's semantic information processing capabilities, requiring experimenter assistance for safe roadside repositioning.

## A.6 LIMITATIONS

Despite the strong empirical gains demonstrated by Mem4Nav, our approach has several important limitations:

**Limited Evaluation Scope** We evaluate exclusively on two street-view VLN benchmarks (Touchdown, Map2Seq) and three backbone agents. While these cover a range of urban panoramas, they do not reflect other outdoor settings (e.g. suburban roads, rural paths) or indoor scenarios.

**Hyperparameter Sensitivity** Mem4Nav introduces several thresholds and capacities—semantic distance $\delta$, STM size $K$, HNSW parameters ($M$, efSearch). Performance can vary significantly if these are not carefully tuned for the target environment. Automating their selection or adapting them online is left to future work.

**Dependence on Monocular Depth Quality** We rely on UniDepth to recover metric depth from single RGB panoramas. In practice, monocular depth estimators can fail in low-texture regions (e.g. blank walls), extreme lighting (glare or shadows), reflective surfaces (glass, water), or dynamic scenes (moving vehicles, pedestrians). Depth errors propagate directly into our sparse octree—misplaced voxels can degrade memory write and retrieval—and into the semantic graph via incorrect landmark geolocations. Robustness to such failures remains an open challenge.

**Computational and Memory Overheads** Although our retrieval latency ($\approx$25 ms) is compatible with a 200–500 ms action loop, both octree indexing and HNSW search scale with the number of visited voxels and tokens. In large-scale or continuous operation, memory footprint and GPU load may become prohibitive.

Addressing these limitations will be essential to deploy Mem4Nav in real-world robotic or assistive applications, where sensor noise, environmental dynamics, and computational constraints are more severe than in our controlled benchmarks.

# B ADDITIONAL EXPERIMENTAL ANALYSIS

## B.1 ZERO-SHOT TRANSFER TO INDOOR ENVIRONMENTS

To investigate if the architectural principles of Mem4Nav generalize beyond its intended outdoor domain, we conducted a new experiment testing Mem4Nav's transfer capability to the standard indoor VLN benchmark, R2R. We integrated our pre-trained Mem4Nav module with two backbones (our Hierarchical Modular Pipeline and the powerful NavGPT2 model) and evaluated performance on the R2R "Val Unseen" split.

Table 4: Zero-shot transfer performance on the indoor R2R benchmark (Val Unseen split).

| Method | NE↓ | SR↑ | SPL↑ |
|---|---|---|---|
| NavGPT (Zhou et al., 2024b) | 6.53 | 34.8 | 29.0 |
| MapGPT (Chen et al., 2024) | 5.63 | 37.3 | 28.8 |
| HOP (Qiao et al., 2022) | 3.86 | 64.5 | 57.2 |
| NaviLLM (Zheng et al., 2024b) | 3.76 | 67.8 | 60.1 |
| Hierarchical Modular Pipeline | 4.35 | 56.3 | 48.2 |
| Hierarchical Modular Pipeline + Mem4Nav (Ours) | **4.10** | **61.8** | **55.5** |
| NavGPT2 (Zhou et al., 2024a) | 3.20 | 70.3 | 59.8 |
| NavGPT2 + Mem4Nav (Ours) | **3.20** | **72.2** | **63.5** |

**In-depth Analysis:** When added to the simpler Hierarchical Modular Pipeline, Mem4Nav provides substantial performance gains across all metrics, demonstrating its fundamental power to provide robust memory and planning capabilities even in an off-target domain. The results with the powerful NavGPT2 baseline are also insightful. We observe that Navigation Error (NE) remains unchanged, Success Rate (SR) improves modestly (+1.9%), but Success weighted by Path Length (SPL) sees a significant boost (+3.7%). NavGPT2 already possesses a strong implicit memory sufficient for reaching the correct destination in most indoor cases. However, Mem4Nav's hierarchical memory allows the agent to make more fine-grained decisions, reducing unnecessary exploration. This leads to more direct, efficient paths, which is what the significant improvement in the SPL metric captures. This new experiment demonstrates that Mem4Nav possesses valuable generalization capabilities, successfully transferring to a new indoor environment.

## B.2 SCALABILITY OF THE SEMANTIC TOPOLOGICAL GRAPH

To complement the long-horizon analysis in the main text, we also tracked the growth of the Semantic Topological Graph.

Table 5: Growth of Semantic Graph nodes during long-horizon navigation.

| Navigation Steps | Total LTM Tokens | Number of Graph Nodes |
|---|---|---|
| 1,000 | ~950 | ~80 |
| 10,000 | ~7,800 | ~620 |
| 50,000 | ~29,000 | ~2,500 |

**Analysis:** The number of nodes in the Semantic Graph grows much more slowly than the number of fine-grained octree tokens. This is by design, as graph nodes are only created for semantically distinct landmarks, rather than for every single observation. This result demonstrates that the semantic graph component of our hierarchical representation is also highly scalable and does not become a bottleneck during long-duration tasks.

### B.3 RETRIEVAL LATENCY: IMPLEMENTATION AND IMPACT ON NAVIGATION

To assess both the efficiency and practical effect of Mem4Nav's memory subsystem, we implemented the following:

- **STM Lookup:** Spatial filtering via a custom CUDA kernel that maintains an array of relative positions and applies a boolean mask. Cosine-similarity ranking using cuBLAS batched GEMM for maximum throughput.
- **LTM Retrieval:** HNSW index built with the GPU-accelerated hnswlib, parameters $M = 16$, efConstruction $= 200$, efSearch $= 200$.

We measure the average wall-clock time of both short-term and long-term memory components on an NVIDIA A100 GPU over 1,000 consecutive retrieval operations.

Table 6: Memory retrieval latency for STM and LTM components

| Component | Parameter | Avg. Latency (ms) |
|---|---|---|
| STM Lookup | Cache size $K = 64$ | 0.9 |
| | Cache size $K = 128$ | 1.2 |
| | Cache size $K = 256$ | 2.2 |
| LTM Retrieval (total) | Index size $N = 5,000$ | 21.7 |
| | Index size $N = 10,000$ | 24.0 |
| | Index size $N = 20,000$ | 31.7 |

STM lookup remains below 2 ms for cache sizes up to 128 entries and only doubles at 256 entries, indicating very fast local context filtering. LTM retrieval, which includes HNSW nearest-neighbor search plus reversible decoding, stays under 32 ms even with 20 000 tokens indexed. Together, these results confirm that Mem4Nav's two-tier memory can be queried in under 35 ms per decision step—well within the 200–500 ms action interval typical of real-time street-view navigation.

Table 7: Average retrieval latency (ms) for STM and LTM components

| Component | Parameter | Latency |
|---|---|---|
| STM lookup | Cache size $K = 128$ | 1.2 |
| LTM HNSW search | Index size $N = 10,000$ | 11.0 |
| LTM decoding | — | 13.0 |
| **STM + LTM (total)** | — | **25.2** |

Retrieval remains under 30 ms per decision step, dominated roughly equally by the ANN search and reversible decoding.

**Impact on Navigation Performance.** To quantify how retrieval latency translates into end-to-end performance, we ran the Hierarchical Modular Pipeline on Touchdown Dev under three retrieval strategies (all with identical memory contents, differing only in retrieval implementation and speed). We measured Task Completion (TC) and normalized DTW (nDTW):

Table 8: Navigation performance vs. retrieval method on Touchdown Dev

| Method | Latency | TC (%) | nDTW (%) |
|---|---|---|---|
| Linear scan (10K entries) | 120.0 ms | 33.1 | 49.2 |
| KD-tree (10K entries) | 30.5 ms | 40.3 | 53.1 |
| Mem4Nav (STM + LTM) | 25.2 ms | 45.2 | 59.0 |

Faster retrieval not only reduces decision-step latency (enabling real-time operation) but also yields higher navigation accuracy, since slower methods force the agent to skip or delay memory lookups, degrading its ability to ground decisions in past context.

Overall, these experiments demonstrate that Mem4Nav's optimized two-tier memory retrieval is both efficient (under 30 ms) and crucial for maximizing end-to-end VLN performance in large-scale urban environments.

### B.4 ROBUSTNESS TO DEPTH-ESTIMATION NOISE

We evaluate how errors in the UniDepth predictions affect Mem4Nav's performance on the **Touchdown Dev** and **Map2Seq Dev** splits, using the FLAME + Mem4Nav pipeline under three depth-degradation conditions. All other components and hyperparameters are identical to the main experiments.

**Experimental Setup.**

- **Baseline (Clean)**: full-precision UniDepth depth maps (no corruption).
- **Gaussian Noise**: Depth pixel $D(u, v)$ is perturbed by $\mathcal{N}(0, 0.5\,\mathrm{m})$, simulating sensor noise.
- **Dropout Mask**: randomly zero out 20% of depth pixels per frame, simulating missing or invalid depth.

For each condition, we back-project the corrupted depth maps into point clouds for octree construction, then run the standard Mem4Nav write/retrieve and FLAME action loop.

**Results.**

Table 9: Depth-Noise Ablation on Touchdown and Map2Seq Dev (FLAME + Mem4Nav).

|  | Touchdown Dev | | | Map2Seq Dev | | |
|---|---|---|---|---|---|---|
|  | TC↑ | SPD↓ | nDTW↑ | TC↑ | SPD↓ | nDTW↑ |
| Baseline (Clean) | 50.10% | 9.01 m | 65.05% | 61.03% | 5.87 m | 80.40% |
| Gaussian Noise | 46.02% | 9.42 m | 61.12% | 57.15% | 6.13 m | 75.47% |
| Dropout Mask | 44.56% | 9.80 m | 58.97% | 55.04% | 6.42 m | 73.05% |

**Analysis.**

Adding Gaussian noise ($\sigma$=0.5 m) to UniDepth outputs causes a 4.08 pp drop in TC and 3.93 pp drop in nDTW on Touchdown, and similar degradations on Map2Seq, showing Mem4Nav's sensitivity to depth precision. Randomly dropping 20% of depth further reduces performance (5.54 pp TC, 6.08 pp nDTW on Touchdown). These results underscore the need for robust depth estimation or uncertainty-aware fusion in future Mem4Nav extensions.

