# OpenReview forum: "Boosting Vision-and-Language Navigation in Urban Environments with a Hierarchical Spatial-Cognition Memory System"
_ICLR.cc/2026/Conference — ICLR 2026 Conference Withdrawn Submission_

### Official Review · Reviewer_DeUF · 2025-10-23

**Soundness:** 3
**Presentation:** 2
**Contribution:** 3
**Rating:** 4
**Confidence:** 4

**Summary:**

This paper introduces Mem4Nav, a hierarchical spatial-cognition memory system designed to enhance Vision-and-Language Navigation (VLN) in urban environments. Mem4Nav integrates a sparse octree for voxel-level indexing and a semantic topology graph for landmark connectivity, coupled with Long-Term Memory (LTM) and Short-Term Memory (STM) modules. The LTM uses reversible Transformer blocks for lossless compression and retrieval, while STM caches recent observations for fast local planning. The system is evaluated across three VLN backbones (modular, LLM-based, and MLLM-based) on Touchdown and Map2Seq benchmarks, showing consistent performance improvements.

**Strengths:**

- **Efficient memory retrieval and scalability**: The dual memory system (LTM + STM) is well-motivated and addresses long-horizon navigation challenges. The use of a sparse octree and semantic graph enables scalable and structured memory retrieval.

- **Strong empirical performance across benchmarks**: Mem4Nav consistently improves navigation metrics across all three VLN backbones (modular, LLM-based, and MLLM-based) on both Touchdown and Map2Seq. Gains include up to 13.3 percentage points in Task Completion, over 12 points in normalized DTW, and reduced final goal distance. These improvements are substantial and consistent, demonstrating the practical value of the proposed memory system.

- **Comprehensive experiments and ablations**: The paper includes detailed ablation studies, hyperparameter sensitivity analysis, and robustness checks, which validate the contribution of each module and support reproducibility.

**Weaknesses:**

1. **The reversible  module shows excessive mathematical wrapping**:
    - In Section 3.2, the authors define two deterministic mappings $F$ and $G$, which together form the forward function R and its inverse $R^{-1}$. This structure is intended to ensure lossless compression and reconstruction. However, the implementation does not appear to enforce strict reversibility. Instead, $R$ and $R^{-1}$ are trained using a cycle-consistency loss, which would be unnecessary if the mappings were truly bijective. This raises doubts about whether the reversible structure is actually used or merely presented for theoretical appeal.
    - If $R^{-1}$ is indeed the inverse of $R$, then decoding the stored token $θ_s^r$ should directly yield the original observation v. The presence of an additional decoder $π_v$ to recover $v$ suggests that $R^{-1}$ alone is insufficient, which contradicts the claim of exact reversibility.
    - It remains unclear whether the reversible Transformer is used as a strict bijective mapping or merely as a learned encoder-decoder pair. Clarifying this distinction is essential for evaluating the soundness of the claims.

2. **Notation inconsistencies**:

    - On page 4, the phrase “Voxel lookup: given a precise coordinate, compute k and fetch $θ_k^r$” uses the symbol $θ_k^r$, which is not defined in the surrounding context.
    -  According to the authors’ definition: $\theta_s^w \leftarrow R(\theta_s^r, v_t)$, and the authors also get: $v_s = R^{-1}(\theta_s^r)$.  Shouldn’t it be: $\theta_s^r, v_t \leftarrow R^{-1}(\theta_s^w)$?
    - In the STM retrieval paragraph, the notation $ i\in C$ is incorrect. $C$ is defined as a set of memory entries $e_i$, not a set of indices $i$.

3. **Excessive formalism**:
    - The paper includes many formulas and symbols that do not contribute meaningfully to the core contributions. This adds cognitive load and detracts from readability. A more concise presentation would improve accessibility.

**Questions:**

1. Why does memory improve VLN performance? In scenarios without repeated path visits, how does the proposed memory mechanism lead to such significant performance gains? Could the authors provide a more detailed explanation?

---

### Official Review · Reviewer_oy35 · 2025-10-27

**Soundness:** 2
**Presentation:** 2
**Contribution:** 2
**Rating:** 4
**Confidence:** 4

**Summary:**

The paper proposes Mem4Nav, a hierarchical spatial-cognition memory system designed to augment Vision-and-Language Navigation (VLN) agents in large-scale urban environments. Mem4Nav integrates a sparse octree for fine-grained 3D indexing and a semantic topological graph for high-level planning, coupled with a dual long-short-term memory mechanism based on a reversible Transformer. The system is evaluated on the Touchdown and Map2Seq benchmarks by integrating it into three distinct backbones (a modular pipeline, VELMA, and FLAME). Results demonstrate substantial improvements in Task Completion and path fidelity (nDTW) across all backbones, claiming new state-of-the-art performance.

**Strengths:**

1. Well-structured hierarchical design integrating geometric and semantic cues.

2. Extensive experiments on multiple backbones and datasets with clear improvements.

3. The paper provides a detailed appendix with implementation specifics, facilitating reproducibility.

**Weaknesses:**

1. The paper repeatedly claims a “lossless reversible Transformer memory.” However, reversibility in neural networks only guarantees gradient reconstruction, not information preservation. Moreover, as the memory is updated over a long sequence of steps, each overwriting the previous state, the overall process is no longer bijective and cannot guarantee lossless recall.

2. Table 3 mainly discusses how different backbones (LLM vs. MLLM) respond to the removal of each module, rather than analyzing the effectiveness of each component.

3. The combination of octree storage and HNSW retrieval likely introduces nontrivial memory and computational overhead, especially in large urban environments.

4. The paper lacks significance tests.

5. The Related Work section omits discussion of episodic or memory systems in embodied AI.

6. The formatting and figures need significant revision.

**Questions:**

See weakness 1 and 3.

---

### Official Review · Reviewer_BZxq · 2025-10-28

**Soundness:** 2
**Presentation:** 2
**Contribution:** 2
**Rating:** 2
**Confidence:** 4

**Summary:**

The paper presents a memory system, Mem4Nav, designed to address the challenge of spatial cognition storage in large-scale Vision-and-Language Navigation (VLN) tasks. This task requires agents to build a comprehensive and accurate understanding of spatial structures while enabling efficient information retrieval for navigation. To overcome the limitations of existing methods in retaining spatial information, the authors introduce a hybrid memory architecture integrating octree-based voxel indexing and topological graph representations, with long-term memory maintained via a reversible transformer. The framework is designed to augment VLN backbones (e.g., FLAME, VELMA, and Hierarchical). Experiments conducted on the Touchdown and Map2Seq datasets demonstrate that Mem4Nav outperforms previous state-of-the-art approaches on several key metrics.

**Strengths:**

1. The paper proposes a hierarchical memory structure that is relevant to the challenges of outdoor VLN.
2. The paper conducts comparative experiments and outperforms existing methods on several key metrics.
3. The code is open-sourced.

**Weaknesses:**

1. The Introduction section provides a weak overview of the field and fails to present a clear motivation. For instance, the narrative does not crisply define what specific capability is missing in existing methods and how Mem4Nav's components uniquely bridge this gap.
2. The authors fail to position this work with previous relevant efforts. Environment map/Memory is widely adopted by recent navigation agents (such as [1,2,3,4]), while the authors do not provide necessary review of these related arts. What is the difference and relations to existing map-/memory-based navigation agents?
[1] Structured Scene Memory for Vision-Language Navigation, CVPR
[2] GridMM: Grid Memory Map for Vision-and-Language Navigation, ICCV
[3] Bird’s-Eye-View Scene Graph for Vision-Language Navigation, ICCV
[4] C-GPT: Empowering Vision-and-Language Navigation with Memory Map and Reasoning Chains
3. The literature review is far from comprehensive, given the huge body of recent efforts in map-/memory-based navigation.
4. In Lines 98–100, the contribution requires revision. Experimental validation is not a contribution and serves only to verify the effectiveness of the proposed method, rather than being an end goal.
5. Figs. 1–3 are unclear; many terms and symbols are not explained. Besides, Fig. 2 duplicates the content of Fig. 1, and the placement of Fig. 2 is inappropriate.
6. The Related Work section should include in-depth discussions of existing scene representation methods and memory mechanisms in VLN.
7. What does "lossless storage" in LTM refer to?
8. The method mainly combines existing techniques (e.g., OctoMap[1], TSGM[2]). What differentiates this method from previous approaches?
[1] OctoMap: An efficient probabilistic 3D mapping framework based on octrees. Autonomous robots 2013.
[2] Topological Semantic Graph Memory for Image-Goal Navigation. In CoRL 2022.
9. The Method section is unnecessarily detailed, with the description of the memory system overloaded by low-level implementation specifics (e.g., Morton codes, read/write tokens). This obscures the high-level algorithmic flow, making the method difficult to understand, verify, and reproduce.
10. The decoupling of long- and short-term memory is a mature design pattern in navigation (e.g., DD-PPO[1], Self-Monitoring Agent[2]). The paper fails to differentiate its LTM-STM mechanism from these prior arts and does not prove its superiority clearly.
[1] DD-PPO: Learning Near-Perfect PointGoal Navigators from 2.5 Billion Frames. In ICLR 2020.
[2] Self-Monitoring Navigation Agent via Auxiliary Progress Estimation. In ICLR 2019.
11. The paper introduces numerous hyperparameters, and most of them (e.g., $\tau$ in Line 305) are set using simple heuristics without clear justification or sensitivity analysis.
12. The paper does not clarify how the bijective mapping of R in Line 252 is rigorously ensured across timesteps.
13. In Line 250, querying the initial observations after multiple time steps involves successive decoding layers. Is this an efficient retrieval mechanism?
14. The paper lacks training details about the reversible transformer.
15. The paper lacks an analysis of the memory system. What is the retrieval accuracy or memory hit rate? How do memory failures influence navigation performance?
16. Table 2 demonstrates that this method is not robust to gaussian depth noise and depth dropout.
17. The paper lacks significance tests and analyses of each component, and of the overall complexity.
18. The Discussion section is overstated, such as "building dynamic, grounded world models" in Line 481.
19. The paper contains many formatting issues (e.g., Lines 749 and 751) and inconsistent mathematical notation (e.g., "$L$").

**Questions:**

See Weakness

---

### Official Review · Reviewer_ewX7 · 2025-11-01

**Soundness:** 3
**Presentation:** 3
**Contribution:** 3
**Rating:** 6
**Confidence:** 3

**Summary:**

This paper introduces Mem4Nav, a framework designed to enhance Vision-Language Navigation (VLN) agents in urban environments. Mem4Nav features a dual-structured spatial 3D map that combines sparse octree indexing with a semantic topology graph augmented by a reversible Transformer-based memory. The authors propose a hybrid memory architecture that integrates a short-term memory (STM) cache for local queries and a retrieval mechanism to balance short- and long-term memories (LTM). Experiments incorporating these hybrid components into three existing VLN backbones on the Touchdown and Map2Seq benchmarks demonstrate consistent performance improvements.

**Strengths:**

Novelty and significance.

__S1__: This paper attempts to address an important problem of quadratic increase of model complexity of Transformer by integrating memory modules.

Quality of experiments.

__S2__: This paper has presented extensive experiments on three VLN backbones on Touchdown and Map2Seq showing its improvements.

__S3__: The authors also conducted further experiments to have a deeper understanding of the proposed method, including STM Policy and Robustness to Depth Noise, ablation study and so forth  with key insights discussed.



Clarity of presentation.

__S4__: The overall presentation is clear and easy to follow with great details.

__S5__: Figures like Fig 2. provides clarity regarding distinction to related works.



__S6__: Code is provided for reproducibility.

**Weaknesses:**

__W1__: __Lack of Text-History Baseline.__ The paper does not include a text-history-only baseline for comparison. Incorporating such a baseline would help differentiate the contribution of the proposed memory module from improvements potentially attributable to textual context alone.

__W2__: __Unclear Motivation for the Reversible Transformer.__ The motivation for using a reversible Transformer is not clearly articulated. While the model employs it to store both the sparse octree and semantic graph, it remains unclear why a reversible design is necessary compared to a standard Transformer. This choice may relate to aligning memory read–write operations or optimizing efficiency, but such reasoning is not explicitly discussed. Clarifying this motivation would strengthen the architectural justification.

__W3__: __Lack of Fine-Grained Analysis Supporting Table 2 Claims.__ Table 2 provides valuable summary findings on the effect of the λ parameter, showing that λ = 0.0 (recency-only) leads to forgetting key landmarks (TC: 48.91%), while λ = 1.0 (frequency-only) results in cognitive inertia and poor adaptation (TC: 48.54%). Although these trends are plausible given the short- vs. long-term memory design, the paper lacks fine-grained metrics to substantiate these claims. It is unclear whether performance drops stem primarily from forgetting landmarks or failing to adapt to new contexts. Including explicit measures such as landmark recognition confidence, cosine similarity in embedding space, or performance across different context-switch frequencies, would provide stronger evidence and deeper insight into the agent’s memory behavior.

__Minor Issues__ (no impact on major contributions):



__M1__: Line 45 missed a space between the word “environments” and “(Schumann”: *“outdoor urban environments(Schumann et al., 2024”*.



__M2__: It would be better to increase the font size in Figure 4 as the text may not be visible when the paper is printed.



__M3__: The order of "Hierarchical Modular Pipeline.” “VELMA (LLM-based).” “FLAME (MLLM-based)” differs from those in Table 3. It would be a good idea to keep the order consistent.

**Questions:**

__Q1__: __Ambiguity in “Dynamic Balancing” of Memories.__ The description that the model *“dynamically balances short- and long-term memories within the agent’s attention”* is somewhat unclear. The hyperparameter λ appears fixed during testing (as shown in Table 2), suggesting the balancing is not dynamic in practice. It would be helpful for the authors to clarify what aspect of the process is dynamic, or to rephrase the statement accordingly.

---

### Note · Authors · 2026-01-03

I have read and agree with the venue's withdrawal policy on behalf of myself and my co-authors.